# Epidemiology of Community-Acquired Sepsis: Data from an E-Sepsis Registry of a Tertiary Care Center in South India

**DOI:** 10.3390/pathogens11111226

**Published:** 2022-10-24

**Authors:** Fabia Edathadathil, Soumya Alex, Preetha Prasanna, Sangita Sudhir, Sabarish Balachandran, Merlin Moni, Vidya Menon, Dipu T. Sathyapalan, Sanjeev Singh

**Affiliations:** 1Department of Infection Control and Epidemiology, Amrita Institute of Medical Sciences and Research Centre, Kochi 682041, Kerala, India; 2Department of Medicine, Amrita Institute of Medical Sciences and Research Centre, Kochi 682041, Kerala, India; 3Department of Medical Administration, Amrita Institute of Medical Sciences and Research Centre, Kochi 682041, Kerala, India; 4Clinical Assistant Professor, Department of Emergency Medicine, Amrita Institute of Medical Sciences and Research Centre, Kochi 682041, Kerala, India; 5Clinical Associate Professor, Department of General Medicine and Division of Infectious Diseases, Amrita Institute of Medical Sciences and Research Centre, Kochi 682041, Kerala, India; 6NYC Health + Hospitals/Lincoln, 234 East 149th Street, Suite 8-20, Bronx, NY 10451, USA; 7Medical Director, Amrita Hospitals and Research Center, Faridabad 121002, Haryana, India

**Keywords:** sepsis, community-acquired sepsis, registry, surviving sepsis campaign

## Abstract

The study aims to characterize community-acquired sepsis patients admitted to our 1300-bedded tertiary care hospital in South India from the Surviving Sepsis Campaign (SSC) guideline-compliant e-sepsis registry stratified by focus of infection. The prospective observational study recruited 1009 adult sepsis patients presenting to the emergency department at the center based on Sepsis-2 criteria for a period of three years. Of the patients, 41% were between 61 and 80 years with a mean age of 57.37 ± 13.5%. A total of 13.5% (136) was under septic shock and in-hospital mortality for the study cohort was 25%. The 3 h and 6 h bundle compliance rates observed were 37% and 49%, respectively, without significant survival benefits. Predictors of mortality among patients with bloodstream infections were septic shock (*p* = 0.01, OR 2.4, 95% CI 1.23–4.79) and neutrophil-to-lymphocyte ratio (*p* = 0.008, OR 1.01, 95% CI 1.009–1.066). The presence of Acinetobacter (*p* = 0.005, OR 4.07, 95% CI 1.37–12.09), Candida non-albicans (*p* = 0.001, OR16.02, 95% CI 3.0–84.2) and septic shock (*p* = 0.071, OR 2.5, 95% CI 0.97–6.6) were significant predictors of mortality in patients with community-acquired pneumonia. The registry has proven to be a key data source detailing regional microbial etiology and clinical outcomes of adult sepsis patients, enabling comprehensive evaluation of regional community-acquired sepsis to tailor institutional sepsis treatment protocols.

## 1. Introduction

Sepsis is a life-threatening medical condition triggered by infection and remains a major challenge for physicians in developing countries like India, where the condition is responsible for 60–80% of lost lives per year [1]. Studies report a prevalence of infections at 40% and severe sepsis or septic shock at 28.3% with a predominance of Gram-negative pathogens in terms of its epidemiology in the country [2]. A recent prospective study has estimated the adult sepsis burden in ICUs to be at 56.4% [3]. Pneumonia, urinary tract infections and bloodstream infections were the common foci of infection in India [4], similar to studies from other developed countries [5]. Several studies on antimicrobial resistance rates using clinical samples revealed an alarmingly high rate of multi-drug-resistant *Klebsiella pneumonia* species (>91%) and 42% of carbapenem-resistant pseudomonas strains [6,7]. Despite the perpetual evolution of evidence-based sepsis management guidelines over time, the scarcity of robust adult sepsis data from India poses challenges in evaluating the applicability of guideline-based recommendations and identifying critical variables associated with early diagnosis and clinical outcomes among sepsis patients in such resource-limited settings.

Surviving Sepsis Campaign (SSC), a collaboration of Society of Critical Care Medicine and the European Society of Intensive Care Medicine, has been setting out protocols and clinical variables to be monitored for identifying ‘at-risk’ patients and time-associated management modalities to manage outcomes [8,9]. The Management of Sepsis in Asia′s Intensive Care Units (MOSAICS) study reported the compliance rates of SSC′s 6 h resuscitation and 24 h management bundles at 6.8% and 8%, respectively. Higher survival rates were noted in sepsis patients with bundle compliance, the mortality rate being 41.1% in cases of non-compliance to management bundles. However, an Indian pediatric study did not observe any association of mortality with 6 h bundle compliance other than severity of illness. A pediatric study had recorded a reduction in mortality with compliance to 1 h bundle proposed as part of the sepsis 2018 update [10,11]. The recent SSC guidelines in 2021 also recommend the administration of antimicrobials within 1 h of sepsis recognition for patients under shock or with increased likelihood of sepsis [9]. Nevertheless, the continuously evolving definitions on sepsis diagnosis and severity over the past decade and the lack of international consensus in triage time and treatment guidelines have posed practical challenges for physicians in sepsis management at bedside at resource-limited settings. 

In order to improve survival and management of sepsis patients, robust data on regional epidemiology and critical clinical variables are essential in addition to early identification of the focus of infection. Moreover, initiatives for timely and effective administration of antibiotics enabling optimized sepsis management, in conjunction with antimicrobial stewardship practices, in the institutional settings as a hospital-level intervention has been highly recommended. Registries are well-established mechanisms for obtaining high quality disease-specific data, generating valuable evidence to study and manage sepsis in our limited resource population for better outcomes. Acknowledging the need to support evidence-based clinical decisions and subsequent protocols tailored to low-middle-income countries (LMIC) with low resource settings, the present study aims to characterize all sepsis patients admitted to our tertiary care hospital in South India by developing a sepsis registry and evaluate the regional spectrum of infections based on focus of infection.

## 2. Methods

Study design and setting: The study was conducted as a prospective observational cohort study. The sepsis registry was designed by a multidisciplinary team at a 1300-bedded, private, tertiary-care hospital in South India which also serves as an apex referral centre for major medical and surgical specialties. Ethics approval for the study was granted by the Institutional Ethics Committee of the hospital. Patients and/or the public were not involved in the design, conduct, reporting or dissemination plans of this research.

Inclusion and exclusion criteria: All adult patients admitted through the emergency department (ED) with a presumed diagnosis of sepsis as per Sepsis-2 criteria following Surviving Sepsis Guidelines (SSG) will be included in the registry and enrolled into the study [12,13]. Patients who acquired sepsis during the course of their stay in hospitals, patients who are less than 18 years of age and pregnant women were excluded.

Development of e-sepsis registry: The clinical data variables to be recorded were identified as per the SSC guidelines and bundle parameters were determined by an expert panel comprising of administrative champions, intensivists, infectious disease physicians, health informaticists and clinical pharmacists. The sepsis registry has six main domains: demographics, sepsis bundles, clinical investigations, microbiological details at 72 h and final patient outcome. Clinical investigations are traced through patient ID and auto-populated into the registry. At 72 h, details of causative pathogens and their antimicrobial resistance status are entered. Clinical isolates exhibiting resistance to at least one antimicrobial agent in three or more antimicrobial drug categories were considered as MDR [14]. SOFA scores were calculated at admission and at 72 h post-admission. The final domain of patient outcomes captured the ICU and hospital stay mortality status along with final diagnosis and cause of death.

Registry data capture workflow: The ED nurses and physicians were trained to identify qualifying sepsis patients as per Sepsis-2 criteria at casualty area [12]. Data entry was designated to the ICU clinical pharmacist for sepsis registry management which proved effective. When a presumed sepsis patient is identified in the ED, an alert is triggered to the clinical pharmacist who visits the patient file and reconciles the data in real time. Data efficiency was guaranteed by daily entry followed by a weekly review by the team for assessing incorrect data type, outliers and data deficiency. 

Statistical analysis: Descriptive statistics were used to summarize the key results. Based on distribution of normality, chi-square and Student *t*-tests or Mann–Whitney U tests were used for categorical and continuous variables appropriately. Significant clinical variables identified through univariate analysis were used in binary logistic regression analysis to identify predictors of outcome. All statistical analysis was performed in SPSS version 17 (IBM, Chicago, IL, USA). *p* < 0.05 was considered to be significant. 

## 3. Result

### 3.1. Baseline Characteristics

Among the 1009 sepsis patients of the registry over a study period of three years since 2014, the predominant age group in the cohort was found to be 61–80 years (41%) and 70% of the sepsis patients were males (Table 1). Septic shock and severe sepsis were observed among 13.5% (136) and 46% (461), respectively, in the study cohort. The in-hospital mortality recorded by the registry was 25%. SOFA scores on admission (ASOFA) were observed to be between two and seven for the majority of patients (60%) and 5% had ASOFA scores above 11. Males were observed to be far more severely ill compared to females, as revealed by a significantly higher mean admission SOFA value of 6.91 ± 3.1 among males to 5.7 ± 3.3 among females (*p* < 0.001). The proportion of severe sepsis (49%) and septic shock patients (15%) was also found to be significantly higher among males in comparison to females, of whom 38% and 10% were found to have severe sepsis and septic shock, respectively (*p* < 0.001).

### 3.2. Clinical and Microbiological Characteristics Based on Focus of Infection

The major focus of infection in the cohort was urinary tract infection (*n* = 364, 36%), followed by pneumonia (*n* = 235, 23%) and 23% of the cohort had bacteremia (*n* = 233). The rest of the identified infective foci including intra-abdominal sources, meningitis, endocarditis and osteomyelitis accounted for 10% among patients with secondary bacteremia. Cultures sent at the time of admission were positive in 572 (57%) of the cohort. Cultures revealed a predominance of Gram-negative bacteria at 60% (383) with a prevalence of Gram-positive bacteria at 21% (135). Fungal infections accounted for 9.7% of the study cohort. The most common pathogens isolated were species of Klebsiella (29%) and *E. coli* (28%). 

The proportion of patients with septic shock was higher among patients with bacteremia (23%) and skin and soft tissue infection (SSI) (20%). Pneumonia and bacteremia had the highest mortality rates at 34% and 32%, respectively (Table 2). Out of 233 patients with bacteremia, 48 patients had primary bacteremia and 185 patients had a bacteremia secondary to an identified focus. Mortality in the primary bacteremia group (18/48,38%) was significantly higher than the secondary bacteremia group (57/185,31%) (*p* < 0.001).

*E. coli* (25%), Enterococcus (22%) and *Klebsiella Pneumonia* (21%) were the most common pathogens among patients with a UTI (Figure 1). A total of 50% of the UTI cases were observed to be harboring MDR pathogens (Figure 2). Among patients with pneumonia, Klebsiella (29%), Acinetobacter (16%) and Pseudomonas (14%) were predominant pathogens with mortality rates of 44% in Acinetobacter (44%) infections. Among cultures for which antimicrobial susceptibility data were available, 38% (*n* = 61) were MDR pathogens. Carbapenem-resistant Enterobacteriaceae (CRE) and ESBL-positive accounted for 21% and 18.3% among MDR pathogens, respectively. 

Predominant pathogens in patients with bloodstream infections were *E. coli* (26%), Klebsiella (21%) and Staphylococcus (14%). Mortality rates for bloodstream infections due to Acinetobacter was 67% and for Streptococcus was at 60%. Monomicrobial infections accounted for 91% (214). Among cultures for which antimicrobial drug resistance results were available, 44% (34) were MDR pathogens. Among patients with skin and soft tissue infections (SSTI), *E. coli* (16%), Klebsiella (14%) and Staphylococcus (14%) were the most common organisms. MDR pathogens accounted for 44% (*n* = 34) among patients with positive cultures in the current admission for sepsis in SSTI.

### 3.3. SSC Bundle Compliance 

The SSC 3 h bundle compliance was 37% (*n* = 371) in the cohort. Among the 26% (*n*= 260) of patients for whom the 6 h bundle was applicable, the compliance to 6 h bundle was observed in 49% (*n* = 128). The mortality was relatively similar in patients compliant (24%) and non-compliant (25%) to 3 h bundle (*p* = 0.32). The proportion of patients with 3 h bundle compliance was significantly low in patients with septic shock at 27%, in comparison to patients without septic shock (*p* = 0.008). No survival benefit was observed among patients with bundle compliance. 

### 3.4. SSC Bundle Compliance and Focus of Infection

Among focus of infection, the 3 h sepsis bundle compliance rates were found to be 35.7%, 37.4%, 34.3% and 35.7% for UTI, pneumonia, bacteremia and SSI, respectively, without a significant association with any focus of infection. The bundle compliance rate was found to be significantly low among UTI patients with septic shock (23.2%) compared to UTI patients without shock (38%) (*p* = 0.035). Similarly, a significantly low 3 h bundle compliance rate was observed among pneumonia patients with septic shock (20%) in comparison to patients without septic shock (40%). No significant association was observed between septic shock and 3 h bundle compliance rates among patients with bacteremia and SSI.

The 6 h sepsis bundle compliance rates measured for patients qualifying for 6 h bundle when mean arterial pressure was <65 in spite of fluid resuscitation among UTI, pneumonia, blood and SSI were estimated to be 46.5%, 49.1%, 50.6% and 54.8%, respectively. No significant association was observed between any focus of infection and 6 h bundle compliance rates. The 6 h compliance rate was observed to be 57%, 47%, 47% and 61.5% among UTI, bacteremia, pneumonia and SSI, respectively. 

### 3.5. Association with Mortality

The association of clinical variables with mortality is depicted in Table 3. Gender pre-disposition was significantly associated with mortality (*p* = 0.02, OR 1.31, 95% CI 1–1.9) with a higher proportion of males expired (27%) than females (21%) in the cohort. Severity of sepsis was significantly associated with mortality (*p* < 0.001) with mortality rates of 8%, 31%, 43% in sepsis, severe sepsis and septic shock categories, respectively.

The mean lactate levels were significantly elevated among patients who expired (2.89 ± 2.5) than survivors (2.25 ± 1.94) (*p* = 0.007). Mortality rates are significantly higher among patients who had a lactate level of 2.5 mmol/L and above (35%) compared to patients who had lactate levels less than 2.5 mmol/L (24%). Patients with Charlson comorbidity index (CCI) scores of three and above had significantly higher mortality at 28% in comparison to patients with CCI scores less than two (16%) (*p* = 0.001, OR 2, 95% 1.42–2.9). A change in SOFA scores from admission to 72 h (ΔSOFA) was significantly associated with outcome (*p* < 0.001), with 35% of patients expired among patients whose ASOFA scores, ranging from 8–11, remained unchanged or elevated for the first 72 h post admission. Presence of CRE was significantly associated with mortality (*p* = 0.024) and septic shock (*p* = 0.001).

### 3.6. Predictors of Mortality among Foci of Infection

Bacteremia (*p* = 0.002, OR 1.64, 95% CI1.19–2.27) and pneumonia (*p* < 0.001, OR 1.8, 95% CI 1.31–2.48) were significantly associated with mortality. Among patients with bacteremia, septic shock (*p* = 0.01, OR 2.4, 95% CI 1.23–4.79) and neutrophil-to-lymphocyte ratio (*p* = 0.008, OR 1.01, 95% CI 1.009–1.066) were significant mortality predictors. Presence of Acinetobacter (*p* = 0.005, OR 4.07, 95% CI 1.37–12.09), Candida non-albicans (*p* = 0.001, OR16.02, 95% CI 3.0–84.2) and septic shock (*p* = 0.071, OR 2.5, 95% CI 0.97–6.6) were significant predictors of mortality in patients with pneumonia. Significant predictors of mortality in the UTI cohort were found to be septic shock (*p* = 0.003, OR 2.9, 95% 1.54–5.66), neutrophil-to-lymphocyte ratio (*p* = 0.049, OR 1.01, 95% CI 1.0–1.034), presence of Candida nonalbicans (*p* = 0.013, OR 2.2, 95% CI 1.18–4.15) and lactate levels of 2.5 and above (*p* = 0.03, OR 3, 95% CI 1.09–8.29).

## 4. Discussion

Our study reports the comprehensive characteristics of 1009 adult sepsis patients recorded in our unique e-sepsis registry. The registry is a key data source for defining the burden of the disease in our community and as an SSC-compliant LMIC database that reflects local epidemiology. It has allowed us to evaluate the etiology of sepsis stratified by focus of infection and determine predictors of mortality, in addition to the evaluation of critical care variables specified in SSC guidelines for its association with mortality.

The relatively lower mortality rate of 25% recorded by our study cohort compared to the previously published studies [15] could be due to the inclusion of sepsis patients admitted through the emergency department, other than hospital-acquired sepsis of ICU focused by other studies. Mortality rates were reportedly low in sepsis patients diagnosed with Sepsis-2 criteria in comparison to Sepsis-3 definitions [16]. Our study also reported the abundance of Gram-negative bacteria across all focus of infection, consistent with other published reports [5,17,18]. The need for broad spectrum empiric Gram-negative coverage for sepsis patients in emergency departments and at casualty, independent of the focus of infection, should be emphasized. MDR etiology was proportionately higher across all foci of infection with UTI harboring the highest MDR isolates at 50%. 

The low 3 h and 6 h bundle compliance rates in our study of 37% and 49% indicates the feasibility of implementing time-bound bundled approaches to be challenging, considering the lack of awareness and training of ED physicians and primary healthcare settings to follow protocolized sepsis care during patient triage. This is further accentuated by barriers such as the challenges in early sepsis detection, ambiguity in sepsis definitions and higher costs for appropriate antibiotic therapy. Unlike insurance-based medical payment models prevalent in developed countries that support antibiotic costs and payouts as per protocolized care, developing countries such as India primarily follow ‘out of pocket’ expenditure models without uniform medical procedure formats and standardized financial structure. Under this context, the potential chances of implementing the recent SSC 2021 guidelines and bundle approach to sepsis care need to be further explored. The absence of significant association of 3 h and 6 h bundles with mortality observed in our cohort provides further credence to the updated SSC guidelines that emphasize a 1 h bundle for patients under shock or who have high chances of harboring sepsis. Moreover, the lack of survival benefits observed with the bundled approach could potentially imply a role for individualized therapy that would need to be studied. 

Despite the prevalence of Enterobacteraciae species in bacteremia, mortality was higher for Gram-positive organisms like streptococci and non-fermenters like Acinetobacter. Acinetobacter was identified as a significant predictor of mortality among patients with pneumonia, though Enterobacteraciae were predominant clinical isolates in all focus of infection. Acinetobacter is known for poor therapeutic management in similar settings and its association with mortality emphasizes the vital contribution of pathogens in sepsis prognosis. The choice of empirical antibiotics needs to be carefully optimized in order to balance the patient survival in LMIC settings of evident epidemiological difference with high MDR burden and the antimicrobial stewardship practices to reduce AMR development. The addition of two empiric antimicrobials having Gram-negative coverage could be suitable for general primary or secondary care LMIC settings with low MDR burden. The high MDR rates as revealed by our registry data could potentially suggest the use of a MDR coverage rather than a double coverage in case of referral tertiary care hospitals with high MDR burden.

SOFA scores at admission have consistently proven to be a reliable predictor of mortality in our study cohort and reported to be superior to SIRS criteria [19]. For patients having ASOFA 11 and below with decreasing SOFA scores over 72 h in our cohort, a relatively high mortality of 25% was observed, contrary to the 6% reported for this category by Ferriera et al. [20]. Neutrophil-to-lymphocyte count ratio (NLR) was observed to be an important predictor of mortality across major foci of infections in our cohort along with septic shock. NLR has previously reported to be associated with poor outcomes and mortality among sepsis patients [21]. 

We believe our sepsis registry enables a comprehensive evaluation of regional data for the integration of sepsis guideline-specific aspects that could be beneficial for the physicians to practice in real-time rather than translating the guidelines as a whole. We recommend tailoring the use of SSC 2021 guidelines to develop institutional specific guidelines optimized to regional microbiological spectrum and antibiograms. The heterogeneous clinical presentation and individual host factors that calls for individualized care in antibiotic use and ICU admission in sepsis supports the prospect of investigating the same in LMICs.

## 5. Limitations

The adult sepsis epidemiology described in our study is based on the data from a single centre, restricting generalizability of results. The data on individual components of 3 h and 6 h bundles were not available. Since the current e-sepsis registry was modeled on SSC 2012 guidelines, the data based on the sepsis 2018 guidelines for recording the 1 h compliance data was unavailable. The pilot e-sepsis registry would be scaled and updated to include the Sepsis-3 definitions and the temporal parameters specific to 1 h bundle.

## 6. Conclusions

The SSC-compliant sepsis registry has proven to be a key data source detailing the regional microbial etiology and subsequent clinical outcomes of adult sepsis patients in our community, enabling comprehensive evaluation of SSC guideline recommendations that can support the formulation of institution-specific sepsis protocols and its feasible integration in clinical practice. 

## Figures and Tables

**Figure 1 pathogens-11-01226-f001:**
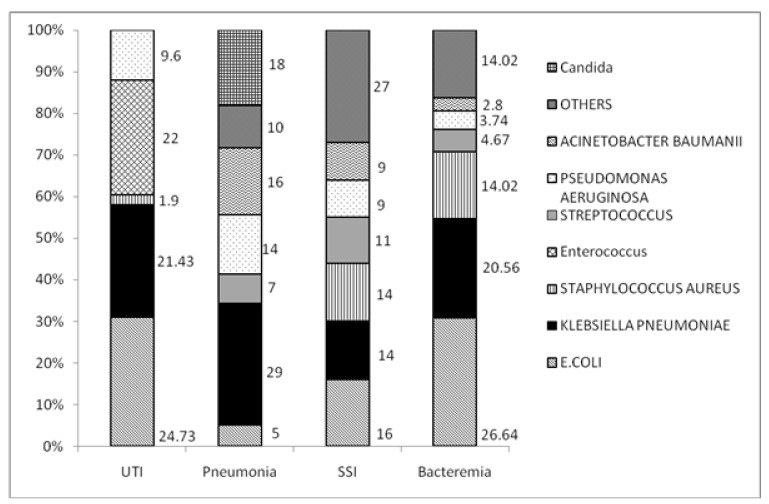
Microbial epidemiology based on focus of infection.

**Figure 2 pathogens-11-01226-f002:**
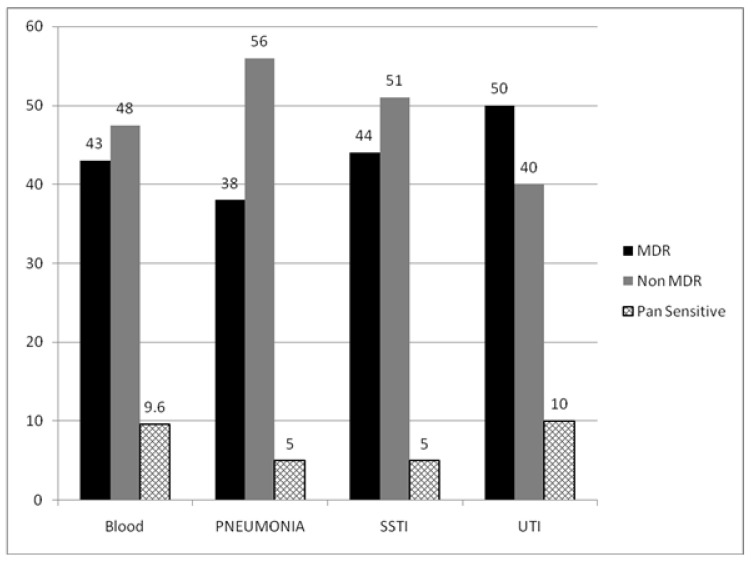
Distribution of drug resistance patterns among focus of infection.

**Table 1 pathogens-11-01226-t001:** Baseline demographics and disease characteristics.

Characteristics	N (%)
Age	
Mean	57.37 ± 16.05
18–40	162 (16)
41–60	371 (37)
61–80	420 (41)
80 and above	56 (6)
Gender	
Females	306 (30%)
Males	703 (70%)
Outcomes	
Alive	760 (75)
Expired	249 (25)
Severity of sepsis	
Sepsis	412 (41)
Severe sepsis	461 (46)
Sepsis with shock	136 (13)
CCI score	
Mean ± SD	3.8 ± 2.4
0 to 2	285 (28)
3 and above	724 (72)
ASOFA scores	
Mean ± SD	6.56 ± 3.2
Asofa 0 to 1	16 (1)
Asofa 2 to 7	606 (60)
Asofa 8 to 11	303 (30)
Asofa above 11	54 (5)
No ABG	30 (3)
Lactate levels	
2.5 and above	252 (24.9)
Less than 2.5	641 (63.5)
Heart rate	
Below 60	15 (1.5)
60–100	488 (48)
Above 100	506 (50)
Temperature	
<100.9	862 (85)
100.9 and greater	143 (14)
Surviving Sepsis Campaign bundle compliance	
3 h bundle compliance	371 (37%)
6 h bundle compliance	128/260 (49%)
Focus of infection	
UTI	364 (36)
Bacteremia	233 (23)
Pneumonia	235 (23)
SSI	112 (11)
Culture positivity	572 (57%)
Type of organism	
Gram-negative	383 (67)
Gram-positive	135 (24)
Fungal	97 (9.6)
Pathogens isolated	
Klebsiella	186 (29)
Ecoli	182 (28)
Enterococcus	99 (15)
Candida non albicans	97 (15)
Pseudomonas	58 (9)
Candida albicans	51 (8)
Staphylococcus aureus	46 (7)
Acinetobacter baumannii	41 (6)
Streptococcus	23 (4)
Proteus	11 (2)
Multidrug resistance	286 (28%)

Abbreviations: CCI: Charlson Comorbidity Index; ASOFA: Admission Sequential Organ Failure Assessment; ABG: Arterial Blood Gas; SSI: Skin and Soft Tissue Infections.

**Table 2 pathogens-11-01226-t002:** Clinical characteristics based on focus of infection.

Characteristics	UTI	Bacteremia	Pneumonia	Skin and Soft Tissue Infections
Mortality	88 (24)	75 (32)	79 (34)	26 (23)
Severity of sepsis				
Sepsis	137 (38)	57 (25)	111 (47)	46 (41)
Severe sepsis	171 (47)	122 (52)	94 (40)	44 (39)
Septic shock	56 (15)	54 (23)	30 (13)	22 (20)
ASOFA score	6.85 ± 3.0	7.47 ± 2.9	6.86 ± 3.2	6.2 ± 2.8
CCI score	4.1 ± 2.3	4.0 ± 2.3	3.6 ± 2.5	4.3 ± 2.3
Monomicrobial infections	262 (72)	214 (91)	113 (48)	43 (38)
Type of organism				
Gram-positive	57 (16)	60 (25)	10 (4)	19 (17)
Gram-negative	163 (45)	165 (71)	80 (34)	45 (40)
Fungal	70 (19)	4 (2)	21 (9)	2 (2)
Antimicrobial drug resistance				
MDR	156 (50)	97 (43)	61 (38)	34 (44)
*CRE*	57 (37)	24 (25)	42 (69)	14 (41)
ESBL positive	56 (36)	32 (33)	13 (21)	14 (41)
Non-MDR	124 (40)	108 (48)	88 (56)	39 (51)
Pan sensitive	33 (10)	22 (10)	8 (5)	4 (5)
CRE	57 (16)	24 (10)	42 (18)	14 (13)
ESBL-positive	56 (15)	32 (13)	13 (6)	14 (13)
SSC bundles				
3 h bundle compliance	130 (36)	80 (34)	88 (37)	40 (36)
6 h bundle compliance	47 (53)	41 (50)	29 (49)	17 (55)

Abbreviations: CCI: Charlson Comorbidity Index; ASOFA: Admission Sequential Organ Failure Assessment; MDR: Multi Drug Resistant; CRE: Carbapenem Resistant Enterobacteriaceae; ESBL: Extended Sectrum Beta-Lactamase; SSC: Surviving Sepsis Campaign.

**Table 3 pathogens-11-01226-t003:** Association with mortality.

Characteristics	Alive	Expired	OR	*p*
Advanced age (≥80 years)	44 (79)	12 (22)	0.82 (0.43–1.49)	0.8
Gender (Males)	517 (73)	186 (27)	1.3 (1–1.9)	0.02
Severity of sepsis				
Sepsis	379 (92)	33 (8)	Ref	
Severe sepsis	317 (69)	144 (31)	5.2 (3.5–7.8)	0.001
Septic shock	64 (47)	72 (53)	12.9 (7.9–21)	0.001
Temperature	99.3 ± 1.3	99.2 ± 1.2	-	0.47
100.9 and above	111 (78)	32 (22)	1.06 (0.61–1.83)	0.83
Less than 100.9	647 (75)	215 (25)	Ref	
Heart rate	100 ± 19.3	106 ± 20.6		<0.001
Below 60	12 (80)	3 (20)	Ref	0.017
60–100	391 (80)	97 (20)	0.992 (0.27–3.5)	0.9
Above 100	357 (71)	97 (20)	1.66 (0.46–6.0)	0.4
Respiratory rate	23.2 ± 6.2	24.4 ± 7.1	-	0.007
Altered mental status	141 (61)	90 (39)	2.48 (1.81–3.14)	<0.001
SBP	132 ± 44	123.3 ± 31	-	0.014
DBP	74 ± 16.8	72 ± 18.7	-	0.13
Pulse pressure	54 ± 21.7	50 ± 22.8	-	0.055
Lactate	2.25 ± 1.94	2.89 ± 2.5	-	0.007
2.5 mmol/L and above	104 (64)	57 (35.4)	1.71 (1.14–2.57)	0.009
Less than 2.5 mmol/L	262 (76)	84 (24)	Ref	
CCI score				
0 to 2	239 (84)	46 (16)	Ref	
3 and above	521 (72)	203 (28)	2 (1.42–2.9)	0.001
ASOFA scores				
Asofa 0 to 1	15(94)	1 (6)	Ref	
Asofa 2 to 7	501 (83)	105 (17)	3.1 (0.41–24.1)	0.27
Asofa 8 to 11	194 (64)	110 (36)	8.5 (1.1–65.2)	0.03
Asofa above 11	22 (42)	31 (59)	21.1 (2.5–172.3)	0.004
No ABG	28 (93)	2 (7)	-	
SSC bundle compliance				
3 h bundle	282 (76)	89 (24)	0.94 (0.7–1.27	0.3
6 h bundle	63 (49)	65 (51)	1.75 (1.07–2.87)	<0.001
Focus of infection				
UTI	276 (76)	88 (24)	0.96 (0.71–1.29)	0.73
Bacteremia	158 (68)	75 (32)	1.64 (1.19–2.27)	0.002
Pneumonia	156 (66)	79 (34)	1.8 (1.31–2.48)	<0.001
SSI	86 (77)	26 (23)	0.91 (0.57–1.45)	0.4

Abbreviations: CCI: Charlson Comorbidity Index; ASOFA: Admission Sequential Organ Failure Assessment; MDR: Multi Drug Resistant; CRE: Carbapenem Resistant Enterobacteriaceae; ESBL: Extended Sectrum Beta-Lactamase; SSC: Surviving Sepsis Campaign.

## Data Availability

The data and material are available with the corresponding author with reasonable request.

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
