# Peer review of "Epidemiology of Community-Acquired Sepsis: Data from an E-Sepsis Registry of a Tertiary Care Center in South India"

_pathogens, 2022, doi:10.3390/pathogens11111226_

Round 1

Reviewer 1 Report

Some typos, i.e., line 42 pnuemonia

Acinetobacter baumanii, change to Acinetobacter baumannii throughout the article.

All tables and figures need a description of abbreviations.

Reviewer 2 Report

Overall adequate.

Needs correction of style and typo. 

Major consideration for changes to table 3.

Reviewer 3 Report

Dear authors

The aim of the manuscript is very interesting and provides information on the characteristics of patients with sepsis in a hospital in a LMIC setting.

These are my comments:

The title reflects the objective of the article.

In the abstract, I consider interesting to specify: in how many years the study is done, what criteria are used to diagnose sepsis: sepsis-2 or sepsis-3 and to include the mean age of the patients. 

In predictors of mortality in the blood stream infections section, Acinetobacter is not significant p = 0.07 and the IC includes number 1. I do not believe it should be included as a significant predictor.

Introduction:

The introduction mentions important aspects of sepsis, such as mortality, increased MDR, in both high income and low middle income countries.

Line 42 please, correct "pnuemonia" to pneumonia.

Line 45 please, specify what MDR means.

Line 46 please, put reference of this paragraph

Line 54 please, put reference of this paragraph. 

Line 76 please, indicate the meaning of LMIC

Methods

Please provide comment on whether informed consent was considered?

I do consider that it is important to specify how sepsis was diagnosed, based on which criteria. Were sepsis-2 or sepsis-3 criteria used?

Line 89 Please explain the meaning of SSG and insert reference to this paragraph. 

Line 103 I believe it is important to define who and where the diagnosis of sepsis is established, and what are the criteria for defining sepsis.

Please explain if all patients are admitted to ICU.

Line 114 Please remove the duplicated paragraph.

Line 142 Please, separate "andskin".

Line 150 Please, separate "pathogenswith".

Results

Line 201 lactate levels appear to be changed, do not match with text.

Line 216 Having an acinetobacterium is considered a predictor of mortality, but this is not statistically significant (p =0.07).

Discussion:

Line 232 The relative low mortality may also be related to the sepsis definition criteria. In the study mortality of severe sepsis 31% and Septic shock 43% acquired in the community are more usual.

Tables and figures

Define acronyms at the bottom of the table.

The tables are not very attractive, I don't know if the presentation could be improved.
